# Acidified Animal Manure Products Combined with a Nitrification Inhibitor Can Serve as a Starter Fertilizer for Maize

**Iria Regueiro** [1], **Peter Siebert** [1,2], **Jingna Liu** [1], **Dorette Müller-Stöver** [1] and **Lars Stoumann Jensen** [1,*]

1. Plant and Soil Science Section, Department of Plant and Environmental Sciences, Faculty of Science, University of Copenhagen, Thorvaldsensvej 40, DK-1871 Frederiksberg, Denmark; iria@plen.ku.dk (I.R.); psi@sagro.dk (P.S.); liu@plen.ku.dk (J.L.); dsst@plen.ku.dk (D.M.-S.)
2. SAGRO, Birk Centerpark 24, 7400 Herning, Denmark
* Correspondence: lsj@plen.ku.dk; Tel.: +45-3533-3470

**Abstract:** There is an urgent need for better management practices regarding livestock farm nutrient imbalances and for finding alternatives to the actual use of mineral fertilizers. Acidification of animal manure is a mitigation practice used to reduce ammonia emissions to the atmospheric environment during manure storage and land application. Acidification modifies manure physicochemical characteristics, among which soluble N and P significantly increase. The main objective of this study was to investigate if acidification and the addition of a nitrification inhibitor to manure and placement of the treated manure close to the seed can stimulate maize growth by enhancing nutrient availability, specially P and consequently plant P uptake, at early development stages without the use of mineral N and P as a starter fertilizer. Raw dairy slurry and solid fractions from dairy slurry and digestate from a biogas plant were acidified to pH 5.5 and applied with or without a nitrification inhibitor (DMPP, 3,4-dimethyl pyrazole phosphate) to maize in a pot experiment, where biomass productivity, nutrient uptake and soil P availability were examined. Acidification increased the water-extractable P fraction of all slurry and digestate organic residues (by 20–61% of total P) and consequently plant P uptake from solid fractions of both slurry and digestate compared to the untreated products (by 47–49%). However, higher plant biomass from acidification alone was only achieved for the slurry solid fraction, while the combination of acidification and DMPP also increased plant biomass in the digestate solids treatment (by 49%). We therefore conclude that the combination of acidification and a nitrification inhibitor can increase the starter fertilizer value of slurry and digestate products sufficiently to make them suitable as a maize starter fertilizer.

**Keywords:** phosphorus; nitrogen; animal manure; manure solids; digestate; DMPP

## 1. Introduction

During the early growth stages of maize, the plant typically has a high nitrogen (N) and phosphorus (P) demand due to a fast plant development rate.

Low soil temperatures have negative effects limiting root development, decreasing the soil volume that can be explored for sources of available P, which results in a reduced root surface area for P uptake. Furthermore, P uptake efficiency per unit root length is decreased at low temperatures [1]. Additionally, P has a low mobility in soil and the speed of diffusion towards the plant root is decreased under cool conditions [2]. This, together with the high demand for exogenous P during early growth stages, can result in plant P deficiency, limiting growth and potential yields in maize crops [3]. An optimal soil or fertilizer P supply during planting and the six-leaf stageis essential to ensure maximum maize

yields [4,5]. A localized P supply close to the roots will therefore increase P availability and uptake and secure early growth [6].

Maize is often grown on livestock farms where basal fertilization is provided through manure, and mineral N and P fertilizers are used to ensure sufficient P availability during early growth stages. This frequently leads to excess P application compared to plant demand, due to a mismatch of the N:P ratio of manures, resulting in a P surplus accumulating in soils causing potential for P loss to the aquatic environment [7]. Additionally, there is a need for closing the anthropogenic P cycle by improving the resource efficiency [8] and an urgent need for better management of the scarce resources of rock P, finding alternatives to the use of mineral fertilizers on livestock farms.

Animal manure is rich in N, P, and organic matter, and has been shown to have a good fertilizer value (and hence economic value) compared to mineral P fertilizers [6]; however, P in manure may not be immediately available for plant uptake shortly after application [9]. Previous experiments investigating the potential of placement of manure next to the seed row to replace mineral fertilizers showed inconsistent results. Schröder et al. [10], Bittman et al. [11] and Pedersen et al. [6] showed very positive effects, such as an increased maize P concentration and maize yield, of manure placement on early maize growth. However, Westerschulte et al. [12] observed a lower plant P uptake from slurry injected close to the maize seed compared to a placed mineral starter P fertilizer, which they attributed to differences in soil P concentration, solubility, and a potentially reduced P availability due to the high slurry pH.

Acidification of manure, by adding sulfuric acid, is used to reduce ammonia ($NH_3$) volatilization due to the induced shift in equilibrium of $NH_3$ and $NH_4^+$ towards the latter [13]. Because of the emission reduction, the nitrogen (N) fertilizer value is increased, as more $NH_4^+$-N is retained in acidified manure and available for the plant. In addition to the effects on N, acidification of slurry also affects other physicochemical manure characteristics [14]. Christensen et al. [15] observed 70% of total P to be dissolved when swine slurry was acidified to pH 5.5. Therefore, by acidifying manure, P will be in a more readily available form for plant uptake, and previous studies have shown how acidified manure has positive effects on early maize growth and P uptake [16].

Thus, animal manure acidification may combine several advantages by increasing the amount of available N and P, while more readily available $NH_4^+$ can enhance P uptake through a local rhizosphere acidifying effect due to the exchange of $NH_4^+$ with $H^+$ and root excretion of $H^+$ when the plant takes up $NH_4^+$ [17]. This acidifying effect in the rhizosphere may potentially dissolve soil P complexes and release readily available forms of P in the rhizosphere, enhancing plant P uptake [18].

Retaining nitrogen as $NH_4^+$ in the upper soil layer for a longer period through the use of a nitrification inhibitor (NI) will often improve N use efficiency and decrease losses to the environment by nitrate leaching or denitrification to $N_2/N_2O$ [19,20]. Additionally, as observed in previous studies, it may also secure the early P supply through rhizosphere acidification and growth of maize when replacing the use of mineral fertilizers [12,21].

The novelty of the current study lies in the combination of the acidification treatment with the addition of a nitrification inhibitor. The aim was to investigate new options to ensure growth stimulation of maize by enhanced nutrient availability at early development stages without the use of mineral N and P as a starter fertilizer. Raw dairy slurry and solid fractions from slurry and from digestate were studied in a pot experiment in relation to their P starter fertilization value. Acidified organic manures with a nitrification inhibitor were examined as possible treatments for increasing plant P availability with regard to their ability to increase P uptake and plant yields during early growth of maize. If these manure treatments prove effective, they could serve as a novel approach to substitute mineral N and P starter fertilizers and thereby reduce nutrient surplus on livestock farms.

We hypothesized that

(i) Acidification of manures will enhance plant P uptake by increasing the water-extractable P (WEP) content in the manures.

(ii) The addition of a NI to acidified manures will increase P availability further.

(iii)   Treated slurry, slurry solids, and digestate solids can replace mineral P starter fertilizers and ensure early growth and optimal biomass productivity of maize.

## 2. Materials and Methods

### 2.1. Organic Manure Characteristics and Acidification

Three types of manure were used in this study: Raw dairy slurry (RS), raw dairy slurry solid fraction (RSSF), and digestate solid fraction (DSF).

Raw dairy slurry was included as a standard reference manure available for crop fertilization on all dairy farms, while the solid fraction obtained by mechanical separation of raw dairy slurry or digestate from an anaerobic digestion plant (for production of biomethane gas) was included because the solids produced by mechanical separation are more concentrated in N and in particular P than their unseparated sources [22], and hence more suitable as a starter fertilizer.

Raw dairy slurry (RS) was collected from a dairy farm in Gelsted, Denmark, in February 2018. The solid fraction of raw dairy slurry (RSSF) was obtained by mechanical separation of RS using a screw press (AGM SB 500 EH, Agrometer a/s, Grindsted, Denmark) and collected from the same farm as RS. The digestate solid fraction (DSF) was collected from the Måbjerg biogas plant in Holstebro, Denmark, after mechanical separation of the anaerobic digester effluent with a decanter centrifuge (UCD 535-00-34, GEA, Westfalia, Germany). The input to the biogas plant includes mainly cow manure (70%), but also pig (20%) and chicken (8–9%) manure, with a small amount of food waste (1–2%) as a co-substrate. Slurry, slurry solids, and digestate solids were stored frozen, from the day after sampling date to usage.

Slurry and solid samples were analyzed at the beginning of the experiment for dry matter (DM), total Nitrogen (TN, Kjeldahl method), ammonium N ($NH_4^+$-N), total P, total potassium (K), and pH as shown in Table 1. Analysis, except pH, was done by a routine chemical lab (OK Laboratorium for Jordbrug, Viborg, Denmark) according to Danish standards for agricultural analyses. The pH of manures was measured by a combined electrode (PHM 210 Meter lab pH meter, Radiometer Medical ApS, Brønshøj, Denmark) in a solution of 2 g manure in 10 mL deionized water.

**Table 1.** Main characteristics of non-acidified raw dairy slurry (RS), raw dairy slurry solid fraction (RSSF), and digestate solid fraction (DSF).

| Organic Manures | DM | TN | $NH_4^+$-N | TP | TK | pH |
|---|---|---|---|---|---|---|
| | (% of WW) | (g kg$^{-1}$ DM) | | | | |
| RS | 8.2 | 50.7 | 22.9 | 10.1 | 20.9 | 8.1 |
| RSSF | 19.3 | 25.1 | 8.9 | 4.5 | 8.5 | 8.6 |
| DSF | 29.6 | 36.0 | 5.7 | 26.2 | 10.5 | 9.1 |

DM: Dry matter, WW: Wet weight, TN: Total nitrogen, TP: Total phosphorus, TK: total potassium.

Prior to the start of the experiments, RS, RSSF, and DSF were acidified in three replicates by placing 20 g samples in 500 mL glass beakers and by gradual additions of 5 μL g$^{-1}$ concentrated sulfuric acid ($H_2SO_4$) with a micropipette until a total amount of 30 μL $H_2SO_4$ was added. After each acid addition, samples were properly mixed by continuous stirring, pH was determined as described above, and titration curves were created (Figure 1) in order to estimate the amount of acid required to lower pH to the desired target value of 5.5 to reduce $NH_3$ loss [14].

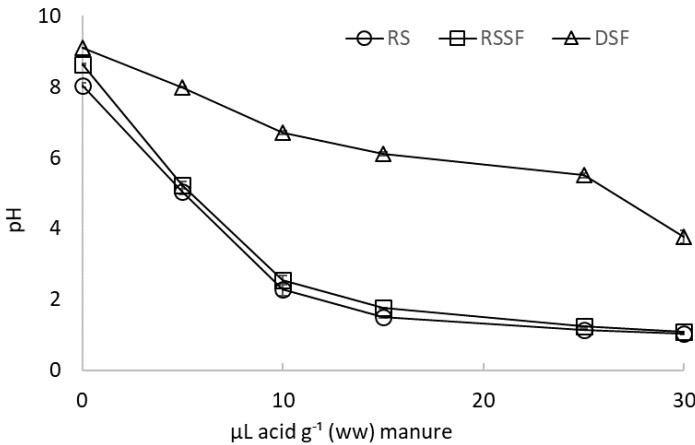

**Figure 1.** Titration curves for raw dairy slurry (RS), dairy slurry solid fraction (RSSF), and digestate solid fraction (DSF) acidified with sulfuric acid, presented as mean values of three replicates. Error bars indicate standard deviation.

### 2.2. Experimental Setup

The soil was collected from the University of Copenhagen's Experimental Farm Højbakkegård in Taastrup, Denmark. The soil is classified as a fine sand (9.8% clay, 10.4% silt, 44.9% fine sand, and 32.3% coarse sand) with a pH (0.01 M $CaCl_2$) of 6.1, 2.1% OM, 35 mg Olsen-P $kg^{-1}$ soil, 104 mg exchangeable K $kg^{-1}$ soil, and 35 mg exchangeable Mg $kg^{-1}$ soil. The soil was mixed before application to pots in a cement blender to ensure a uniform water content and sieved through a 4 mm sieve. Standard solutions of relevant essential macro- and micro-nutrients except N and P were added to the soil (i.e., adding K 110, S 29, Ca 20, Mg 20, Cu 0.5, Zn 1.2, Mo 0.07, Fe 0.3, B 0.2, Mn 3.4 mg $kg^{-1}$ soil).

The maize pot trial was conducted in a climate-controlled chamber with 16 h day temperature of 17 °C and 8 h night temperature of 13 °C. The light intensity was 500 µmol $m^{-2}$ $s^{-1}$. The pots were PVC tubes of 28 cm height and 10 cm diameter (total volume 2.2 L) filled with 2.5 kg (dry weight) soil (further details below), each grown with one maize plant. Pots were completely randomly distributed in the climate chamber and rotated every 3–4 days to minimize possible position effects.

A preliminary pot experiment was performed to establish the plant response to increasing dosages of N and P fertilizer. Mineral fertilizer treatments were tested in the range 0–225 mg N $kg^{-1}$ soil (as $NH_4NO_3$) and 0–80 mg P $kg^{-1}$ soil (as finely ground triplesuperphosphate, TSP, 20.8% P). All treatments received 40 mg of the total N applied and all P was applied as a starter fertilizer in a layer at 10–12 cm depth. The remaining amount of N was applied as topdressing 11 days after sowing (DAS) when maize plants had 1–2 leaves. The preliminary trial was harvested 28 DAS and established that 40 mg P $kg^{-1}$ soil and 150 mg N $kg^{-1}$ soil were sufficient rates to ensure maximum biomass production (no further increase at higher application rates).

In the main experiment, the three organic manures were applied to the pots at rates to achieve 40 mg P applied $kg^{-1}$ soil (Table 2), alone or after acidifying to pH 5.5 with $H_2SO_4.$

In additional treatments, a nitrification inhibitor (NI) (Vizura, BASF) was added to the raw slurry and to each of the acidified products. The Vizura NI contains the active ingredient 3,4-dimethyl pyrazole phosphate (DMPP), which had a concentration of 167 g active ingredient $L^{-1}$ Vizura. The NI was applied at a dosage of 1% of the total manure N applied (i.e., 1 L per 100 kg N applied). Reference treatments receiving increasing amounts of mineral N ($NH_4NO_3$) and P (TSP) fertilizer were additionally set up (see Table 2). A negative control treatment without external addition of N and P fertilizer (N0P0) was also established. Each treatment was carried out in four replicates, resulting in a total of 60 pots.

The soil was added to the PVC tubes in three steps: 1. Addition of first soil layer, 2. Addition of fertilizer in a layer, 3. Addition of the remaining soil. The bottom and top layer corresponded to approximately half of the total soil amount (1.25 kg dry soil), and during the filling process, soil compacting was performed to

reach a bulk density of approximately 1.5 g DM cm$^{-3}$, corresponding to the typical field density of this soil type. This ensured a depth of 10–12 cm from the soil surface to the fertilizer layer [6]. The mineral fertilizer treatments received 40 mg N kg$^{-1}$ soil and all the P required for the treatment in the fertilizer layer.

**Table 2.** Nutrients applied to each pot with the application of the different manures and mineral fertilizers. Organic fertilizers were applied at different weights to reach the same level of 40 mg P added kg$^{-1}$ soil (pot = 2.5 kg of dry soil).

| Treatment * | Manure Applied | TP Applied | WEP Applied | TN Applied in Manure | Mineral N Applied in Manure | Top Dressed NH$_4$NO$_3$-N | TN Applied |
|---|---|---|---|---|---|---|---|
| | (g kg$^{-1}$ soil) | (mg kg$^{-1}$ soil) | | | | | |
| RS | 48.8 | 40 | 8.4 | 202 | 91.2 | 150 | 352 |
| aRS | 48.8 | 40 | 26.8 | 202 | 91.2 | 150 | 352 |
| niRS | 48.8 | 40 | 26.8 | 202 | 91.2 | 150 | 352 |
| aniRS | 48.8 | 40 | 26.8 | 202 | 91.2 | 150 | 352 |
| RSSF | 46.4 | 40 | 24.8 | 225 | 80 | 150 | 375 |
| aRSSF | 46.4 | 40 | 32.8 | 225 | 80 | 150 | 375 |
| aniRSSF | 46.4 | 40 | 32.8 | 225 | 80 | 150 | 375 |
| DSF | 5.16 | 40 | 4.4 | 55 | 8.63 | 150 | 205 |
| aDSF | 5.16 | 40 | 28.8 | 55 | 8.63 | 150 | 205 |
| aniDSF | 5.16 | 40 | 28.8 | 55 | 8.63 | 150 | 205 |
| N0P0 | - | 0 | - | - | 0 | 0 | 0 |
| N150P0 | - | 0 | - | - | 40 | 110 | 150 |
| N150P20 | - | 20 | - | - | 40 | 110 | 150 |
| N150P40 | - | 40 | - | - | 40 | 110 | 150 |
| N225P40 | - | 40 | - | - | 40 | 185 | 225 |

* RS: Raw slurry; RSSF: Raw slurry solid fraction; DSF: Digestate solid fraction; a: Acidified; ni: Nitrification inhibitor; ani: acid+ni; N0P0: Control, no N and no P; N150P0: Mineral N without P; N150P20: Mineral N with low P; N150P40: Mineral N with high P; N225P40: High mineral N with high P. TN: Total nitrogen, WEP: Water Extractable Phosphorus, TP: Total phosphorus.

Two maize seeds of the variety Ambition (Limagrain A/S) with a seed weight of 270–320 mg seed$^{-1}$ were sown in each pot at 4 cm depth from the surface. When the seeds had germinated, one of the seedlings was removed. A top dressing of mineral N (NH$_4$NO$_3$ solution) was applied 13 days after sowing (DAS), to ensure sufficient N supply for plant growth (Table 2).

Pots were irrigated every 2 days during the duration of the experiment to 65% of the pot water holding capacity (WHC), corresponding to ca. 60% water-filled pore space. The WHC of a pot containing 2.5 kg soil was determined by water saturation and subsequent free drainage for 24 h, and was found to be 26% of the dry soil weight.

## 2.3. Analytical Methods

The pH of manures was measured by a combined electrode (PHM 210 Meter lab pH meter, Radiometer Medical ApS, Brønshøj, Denmark) in a solution of 2 g manure in 10 mL deionized water. The remaining manure analyses were done by a routine chemical lab (OK Laboratorium for Jordbrug, Viborg, Denmark) according to Danish standards for agricultural analyses. Ammonium and nitrate contents of manures were analyzed by the extraction of wet solids in 1 M KCl (1:20 g DM:mL vol) by 60 min shaking in an end-over-end shaker. Extracts were centrifuged, filtered through filter papers (Whatman no. 5) and analyzed by flow injection analysis (FIAstar 5000, Foss, Sweden).

Aboveground biomass was harvested 35 DAS by cutting the maize plants 1.5 cm above the soil surface. Roots were carefully washed to remove adhering soil, and shoot and root dry matter yield was determined after drying at 60 °C for 48 h. The dry shoot biomass was finely milled, and the total N content of plant samples was analyzed by elemental analysis (Vario macro cube, Elementar Analysensysteme GmbH, Germany). Total P of plant samples was determined by microwave digestion with nitric acid and hydrogen peroxide and measurement by flow injection analysis (FIAstar 5000, Foss, Sweden).

A soil sample was taken after harvest with an auger vertically from the soil surface to 18 cm depth to include the fertilizer layer. The soil was analyzed for water-extractable P (WEP), NH$_4^+$, and NO$_3^-$ content. WEP was determined by a 1:60 (soil g DM:mL vol) extraction in milli-Q water by

60 min shaking in an end-over-end shaker. The extracts were centrifuged, filtered through Whatman no. 5 filter papers and analyzed by flow injection analysis (FIAstar 5000, Foss, Sweden). $NH_4^+$ and $NO_3^-$ content was determined after extraction in 1 M KCl (1:4 soil g DM:mL vol) by shaking for 60 min. Extracts were centrifuged, filtered through filter papers (Whatman no. 5) and analyzed by flow injection analysis (FIAstar 5000, Foss, Sweden).

### 2.4. Data Calculation and Statistical Analysis

Total inorganic N was calculated as the sum of $NH_4^+$-N and $NO_3^-$-N. Total P and N uptake by plants was calculated by multiplying shoot dry matter biomass yields (g DM pot$^{-1}$) with P and N concentration (mg g$^{-1}$ DM) of harvested biomass, respectively.

Statistical analyses were conducted using ANOVA and the Tukey test at the 0.05 significance level for comparisons. For all ANOVAs, the assumption of the homogenous variance of different groups was checked with Levene's test and the assumption of normality was tested using the Kolmogorov–Smirnov test. All statistical analyses were conducted using IBM SPSS Statistics 20.

## 3. Results

### 3.1. Acid Requirement to Decrease Manure pH

The amount of sulfuric acid required to decrease the pH to 5.5 in DSF (25 μL $H_2SO_4$ g$^{-1}$ ww) was notably higher than the amount required in RS and RSSF (4.5 μL $H_2SO_4$ g$^{-1}$ and 4.9 μL $H_2SO_4$ g$^{-1}$ ww, respectively), clearly indicating that DSF has a higher buffer capacity (Figure 1) in the interval from original to target pH (5.5). The highest buffering capacity for DSF was observed between pH 6.7 and pH 5.5, while for RS and RSSF, buffer capacity was highest between pH 2.5 and pH 1.5 (Figure 1).

### 3.2. Phosphorus Solubilization by Acidification

Acidification of all three manure products significantly increased the WEP concentration compared to non-acidified materials (Figure 2). Prior to the acidification treatment, WEP represented 21%, 62%, and 11% of the total P in RS, RSSF and DSF, respectively, and after the acidification, WEP increased to 67, 82, and 72% of the total P in the same samples, respectively. Therefore, aDSF showed the highest increase in WEP of 61% of total P, or more than a six-doubling of WEP.

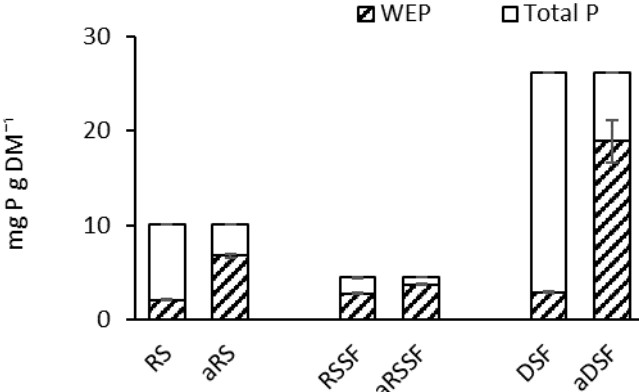

**Figure 2.** Water extractable phosphorus (WEP) in non-acidified organic products: Raw dairy slurry (RS), raw dairy slurry solid fraction (RSSF) and digestate solid fraction (DSF), and in acidified organic products: acidified dairy slurry (aRS), acidified dairy slurry solid fraction (aRSSF) and acidified digestate solid fraction (aDSF).

Addition of the NI did not affect WEP (data not shown) and although the NI (DMPP) contains a phosphate group, at an addition rate of 1% of slurry N content, this is negligible compared to the natural manure phosphate content.

## 3.3. Pot Trial: Plant Biomass Yield and Nutrient Uptake

Plants receiving mineral N and P fertilizer treatments showed a significant plant biomass yield increase proportional to the increase in N and P supplied (Figure 3, right side). All pots receiving untreated or treated organic manures had a plant biomass yield at 35 DAS equal to pots receiving the mineral fertilizer with highest N and P applied N225P40 (Figure 3, left side and center).

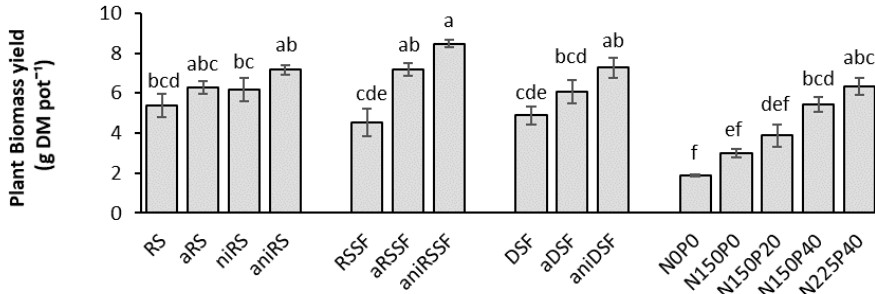

**Figure 3.** Plant biomass yield for untreated and acidified organic products and reference mineral fertilizers 35 days after sowing (DAS) as mean values of 4 replicates. RS: Raw slurry; RSSF: Raw slurry solid fraction; DSF: Digestate solid fraction; a: Acidified; ni: Nitrification inhibitor; ani: acid+ni; NxPy: mineral N and P reference fertilizer at x mg N and y mg P kg$^{-1}$ soil. Error bars represent standard error of the mean (*n* = 4). Treatments with different letters (across all levels) are significantly different (*p* < 0.05) according to the Tukey test.

Acidification alone did not significantly increase plant biomass yield when aRS and aDSF were applied compared to non-acidified manures; however, a significantly higher (58%) plant biomass yield was observed with the application of aRSSF compared to RSSF. When acidification and the NI were combined, plant biomass yield in the treatments receiving the raw slurry solid fraction did not significantly increase further compared to aRSSF; however, a 49% increase in plant biomass was observed in aniDSF compared to DSF.

The plant biomass yield increased in our study when aniDSF was applied, and this was as well reflected in the root:shoot ratio (Figure 4) as similar trends were observed in all ratios obtained, except for RSSF. Values obtained for DM yield of roots (values not shown) were in accordance to values obtained for DM yield of shoot biomass (Figure 3) in each treatment. When the untreated solid fraction of slurry (RSSF) was applied, a significantly higher root–shoot ratio was obtained compared to aRSSF and aniRSSF (Figure 4).

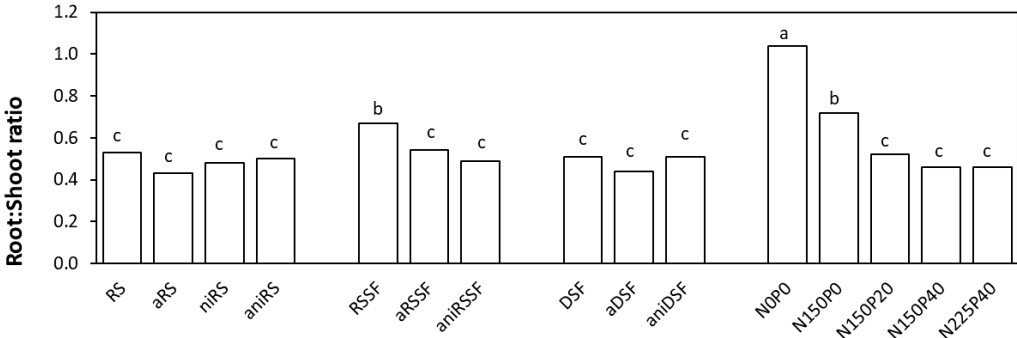

**Figure 4.** Root:Shoot ratio for untreated and acidified organic products and reference mineral fertilizers 35 days after sowing (DAS) as mean values of 4 replicates. RS: Raw slurry; RSSF: Raw slurry solid fraction; DSF: Digestate solid fraction; a: Acidified; ni: Nitrification inhibitor; ani: acid+ni; NxPy: mineral N and P reference fertilizer at x mg N and y mg P kg$^{-1}$ soil. Treatments with different letters (across all levels) are significantly different (*p* < 0.05) according to the Tukey test.

All treatments resulted in a significant increase of plant N uptake compared to the control treatment N0P0. As in the preliminary trial, the increase from 150 mg mineral N kg$^{-1}$ soil to 225 mg mineral N kg$^{-1}$ soil did not result in a further increase in plant N uptake. In the treatments receiving treated manure products, equal or higher values of N uptake, even if not statistically different, were obtained compared to the control receiving the highest amount of mineral N and P fertilizer (N225P40) (Figure 5).

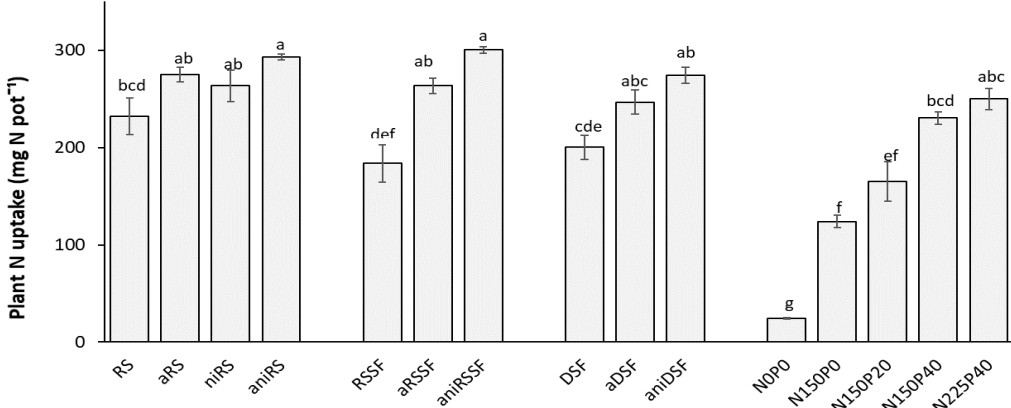

**Figure 5.** Total N uptake for untreated and acidified organic products and reference mineral fertilizers 35 days after sowing (DAS) as mean values of 4 replicates. RS: Raw slurry; RSSF: Raw slurry solid fraction; DSF: Digestate solid fraction; a: Acidified; ni: Nitrification inhibitor; ani: acid+ni; NxPy: mineral N and P reference fertilizer at x mg N and y mg P kg$^{-1}$ soil. Error bars represent standard error of the mean ($n = 4$). Treatments with different letters (across all levels) are significantly different ($p < 0.05$) according to the Tukey test.

Acidification significantly increased N uptake when aRSSF was applied compared to RSSF. When the NI was added to acidified manures, the N uptake significantly increased with the application of all manures compared to the untreated ones (Figure 5).

The mineral fertilization treatments showed a gradual increase in plant P uptake with increasing rates of P fertilizer applied. All manure applications resulted in a significant increase in plant P uptake compared to the control N0P0 and all led to a similar P uptake as in the mineral treatment receiving the highest P and N fertilizer (N225P40).

The plant P uptake significantly increased when acidification was applied to solid fractions of raw and digested slurry. Thus, increases of 49% and 47% were obtained when aRSSF and aDSF were applied to pots, respectively, compared to RSSF and DSF. The uptake of P from these slurries was further increased when NI was added to acidified samples, with increases of 83% and 81% compared to the untreated materials when aniRSSF and aniDSF were applied, respectively (Figure 6).

Acidification did not have a significant impact on the P uptake when RS was applied; however, the application of aniRS showed a significant increase in P uptake, compared to RS application. For all three organic products to which both acidification and the NI was applied, values for P uptake were significantly higher compared to values obtained when the highest rate of mineral fertilizer N225P40 was applied (Figure 6).

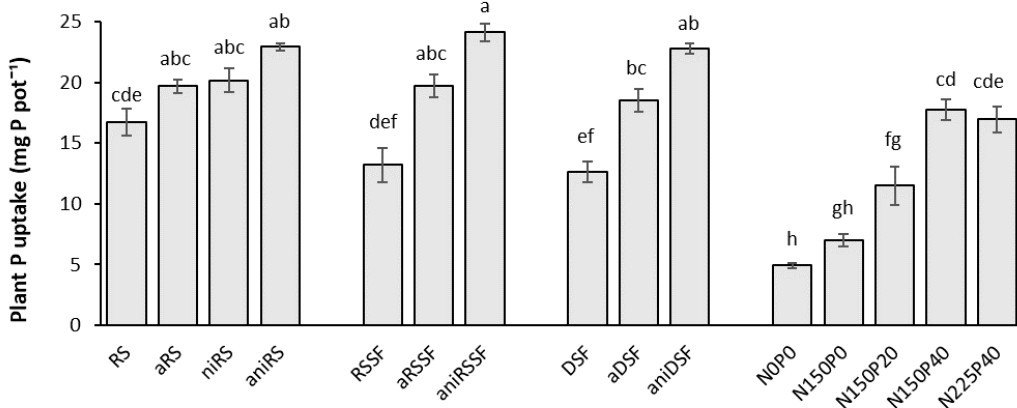

**Figure 6.** Total P uptake for untreated and acidified organic products and reference mineral fertilizers 35 DAS as mean values of 4 replicates. RS: Raw slurry; RSSF: Raw slurry solid fraction; DSF: Digestate solid fraction; a: Acidified; ni: Nitrification inhibitor; ani: acid+ni; NxPy: mineral N and P reference fertilizer at x mg N and y mg P $kg^{-1}$ soil. Error bars represent standard error of the mean (*n* = 4). Treatments with different letters (across all levels) are significantly different (*p* < 0.05) according to the Tukey test.

### 3.4. Residual Mineral N and WEP in Soil after Harvest

The WEP in soil at harvest did not differ greatly among the different organic products applied and similar values were obtained compared to the highest amount of mineral fertilizer applied (Figure 7). When RSSF was used, values of WEP in soil were slightly (but not significantly) higher compared to values obtained when aRSSF and aniRSSF were used.

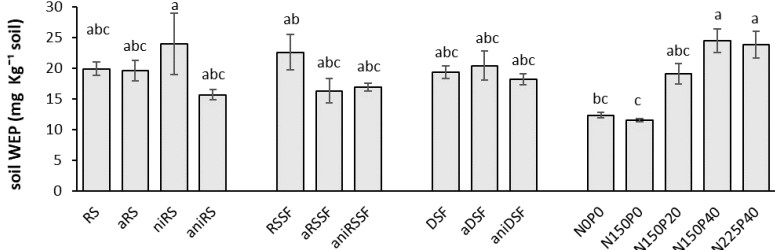

**Figure 7.** Postharvest WEP in soils applied with untreated and acidified slurries and reference mineral fertilizers 35 DAS as mean values of 4 replicates. RS: Raw slurry; RSSF: Raw slurry solid fraction; DSF: Digestate solid fraction; a: Acidified; ni: Nitrification inhibitor; ani: acid+ni; NxPy: mineral N and P reference fertilizer at x mg N and y mg P $kg^{-1}$ soil. Error bars represent standard error of the mean (*n* = 4). Treatments with different letters (across all levels) are significantly different (*p* < 0.05) according to the Tukey test.

The total mineral N content in soil at harvest varied greatly among the different organic manures applied. When solid fractions from raw slurry (RSSF) and digestate (DSF) and their respective treatments were applied, values of total inorganic N remaining in the soil after harvest were significantly lower compared to RS and the corresponding RS treatments (Figure 8).

Acidified manures applied to pots did not have a significant effect on the remaining mineral N in the soil after harvest. However, the application of the NI alone and in combination with acidification to raw slurry (niRS and aniRS) significantly increased the fraction of $NH_4^+$-N in the soil sampled at harvest. When aniRSSF and aniDSF were applied, the $NH_4^+$-N fraction showed similar values and the $NO_3$-N fraction significantly decreased compared to the application of the same non-treated manures (Figure 8).

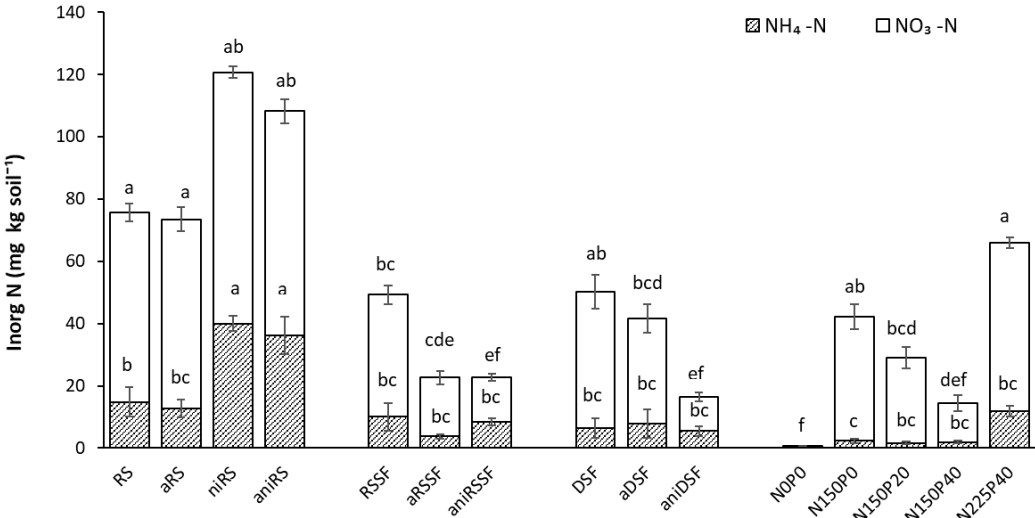

**Figure 8.** Inorganic N ($NO_3$-N + $NH_4$-N) in soils applied with untreated and acidified products and reference mineral fertilizers 35 DAS as mean values of 4 replicates. RS: Raw slurry; RSSF: Raw slurry solid fraction; DSF: Digestate solid fraction; a: Acidified; ni: Nitrification inhibitor; ani: acid+ni; NxPy: mineral N and P reference fertilizer at x mg N and y mg P $kg^{-1}$ soil. Error bars represent standard error of the mean (*n* = 4). Treatments with different letters, for each mineral N type and across all levels, are significantly different (*p* < 0.05) according to the Tukey test.

## 4. Discussion

### 4.1. Buffer Capacity of Manures, Acid Used, and Effects on WEP

The amount of sulfuric acid required to decrease the pH in slurries depends on the buffer capacity, which is mainly affected by the content of buffer components, i.e., ammoniacal N, carbonates, phosphates, and volatile fatty acids (VFA). The raw dairy slurry used in our study appears to be of normal composition. Undigested cattle slurry was reported to require 2.72 µL $H_2SO_4$ $g^{-1}$ (ww) or 54 µL $H_2SO_4$ g $DM^{-1}$ [23], which is in accordance with the 4.5 µL $H_2SO_4$ $g^{-1}$ (ww) or 54.9 µL $H_2SO_4$ g $DM^{-1}$ used to decrease the pH to 5.5 of raw dairy slurry (RS) in our study.

Volatile fatty acids are reduced during biogas production, and their contribution to the buffer capacity in digestates is therefore negligible [24]. However, in accordance with the results of this study, digestates have normally a higher buffer capacity than fresh manures, as alkalinity increases with the $CO_2$ produced during the anaerobic process [25]. The solid fraction from digestate has normally an additional increase in buffer capacity due to most of the Ca and Mg phosphates ending up in this fraction after separation [26].

A higher amount of acid than in our study was required in another study (66.2 µL $H_2SO_4$ g $DM^{-1}$) to decrease the pH to 5.5 in solid fractions from digestate [27]. In their study, digestate solids had a similar DM, N, and P content as in our study; however, the $NH_4^+$ content was higher, perhaps indicating that ammoniacal N had more influence on the buffer capacity.

Untreated organic manures had different initial P concentrations and the digestate solid faction showed the highest P concentration of all treated products, followed by RS and RSSF (Table 1) and the same decreasing order was observed in relation to acid required to decrease pH (Figure 1). Even though RSSF had a higher initial pH than RS, the amount of acid required to decrease the pH in RSSF was lower than in RS, indicating that the notably higher P concentration could be the main responsible for higher acid requirement, with precipitated Ca and Mg phosphates likely being the main variables affecting buffer capacity.

Phosphorus concentration in solid fractions is normally high, as P tends to form precipitates, be part of or adhere to particles in slurry. However, the transfer of P to the solid faction varies depending

on the particle size distribution of the slurry and the separation technique used. Mechanical separation by screw press forces small particles through the filter pores and a large part of the small particles ends up in the liquid fraction [28,29], which could explain the low P content in RSSF. Most of the P present in digestates is contained in the solid fraction after decanting centrifugation applied in most biogas plants, and this is likely because this separation technique is able to retain smaller particles [22] explaining the high P concentration in DSF (Table 1).

Water extractable phosphorus (WEP) is a good indicator for the P plant availability in manures and consists of both inorganic and organic fractions of P [30]. However, previous studies have reported 75% of WEP in dairy slurry to be in inorganic form [31]. Struvite and dicalcium-phosphate are abundant in animal manure. Previous studies have shown high increases in soluble phosphorus concentration when decreasing pH to 5.5 with sulfuric acid [32] and it has been attributed to struvite dissolution [33], but also to partial hydrolysis of less soluble fractions of organic P to more labile forms [34], explaining the WEP increase.

The proportion of WEP of total P in RSSF was higher than in RS, and the WEP increase observed after acidification was lower in aRSSF (20%) compared to aRS (46%). Similar results were observed by Pedersen et al. (2017), where cattle slurry with an initial WEP concentration of 69% of total P and acidified to pH 5.5, showed an increase of 6%. The low WEP increase in RSSF is most likely due to the higher initial concentration of WEP and a lower proportion of Ca and Mg phosphates that had been associated to smaller particles in the slurry and were therefore not retained in the solid fraction.

## 4.2. Treatments Effect on Nutrient Uptake and Plant Growth

All pots receiving untreated organic manures showed similar biomass yields and P uptake, in spite of the low WEP and $NH_4$ addition in DSF compared to RS and RSSF (Table 2). A better nutrient uptake could have been expected when RSSF was applied since a higher N and WEP content was applied (Table 2) but possibly some N immobilization occurred due to the high amount of organic matter added. As previously mentioned, when RSSF was applied, a significantly higher root:shoot ratio was obtained compared to aRSSF and aniRSSF (Figure 4) possibly indicating the nutrient deficiency during shoot development when RSSF was applied, which did not occur with the acidification treatment and with its combination with the NI. Additionally, the low P uptake and biomass yield when RSSF was applied is supported by the slightly higher (but not significantly) WEP value remaining in the soil after harvest, confirming the nutrient deficiency when RSSF was applied compared to values obtained when aRSSF and aniRSSF were used (Figure 7). Acidification alone increased the plant N uptake only from the solid fraction from raw slurry (aRSSF) compared to the untreated RSSF which could be as well explained by the nutrient deficiency that occurred when RSSF was applied.

Acidification increased P uptake from both acidified solid fractions (aRSSF and aDSF) compared to the untreated samples. Among these solid fractions, the main difference lays in the initial P content (Table 1) and their percentage of WEP of total P (Figure 2). Since the application rates in the pot experiments were based on total P addition, the amount of total N and the corresponding $NH_4$ with aRSSF was higher than with aDSF (Table 2). However this difference in N applied did not affect the P uptake.

Previous studies have shown that $NH_4^+$ enhances root growth and promotes acidification of the rhizosphere with positive effects on the uptake of P [14] and this, together with the increase in soluble P when acidification was performed may have been the main reason for the increase in P uptake when acidified solid fractions were applied (aRSSF and aDSF) compared to untreated solid fractions (RSSF and DSF). A significant increase in biomass yield however was only observed when aRSSF was applied compared to RSSF and this increase may have been influenced by the high percentage of WEP in total P (82%) in aRSSF (Figure 2) promoting a higher shoot biomass yield.

The balance between mineralization and immobilization is influenced by the C:N ratio of the degradable OM in manures [35] but other parameters, such as the $NH_4$-N:TN ratio or the easily degradable or recalcitrant C present in the slurry might also influence the net N mineralization.

The digested manure (DSF) had the highest P concentration compared to raw slurry and its solid fraction. However, $NH_4$-N content in DSF is the lowest from all three organic products (Table 1), which indicates that ammonium N may have played a role in the uptake of P. This was supported by the significantly lower P concentration obtained in the shoot biomass when DSF treatment was applied.

The plant biomass yield increase promoted when aniRSSF and aniDSF were applied may be explained on the one hand by the high proportion of WEP in total P, as well as by a higher OM content in these solid fractions compared to raw slurry. The use of a NI is likely to enhance the interaction between the inorganic N and P in the rhizosphere, by preventing nitrification of the ammonium in the manure [6]. The addition of a NI promoted a higher $NH_4^+$ uptake, which consequently improved the P availability as the uptake of N observed is correlated with the P uptake and directly linked to the plant biomass yield obtained. This biomass increase when aniRSSF and aniDSF were applied is as well supported by the significantly lower $NO_3^-$ content found in the soil after harvest when these treatments were applied (Figure 8). In environments prone to leaching, the addition of a nitrification inhibitor to the starter fertilizer may therefore also have a reducing effect on nitrate losses [36].

Thus, the combination of acidification and a nitrification inhibitor could enhance the starter fertilizer effect of manure products. The addition of $H_2SO_4$ has furthermore been shown to be beneficial for plant Sulphur (S) supply, but may on the long run lead to excess amounts of S added to soil [37]. This may, however, be less problematic for materials derived from anaerobic digestion, as a large part of the S contained in the input material is lost during the digestion process [38]. Although the use of nitrification inhibitors is promoted as a measure to reduce adverse environmental effects of nitrate leaching and $N_2O$ emissions [39], recent studies on their ecotoxicology under field conditions are rare. Some investigations suggest that certain substances may have a negative effect on ammonium-sensitive crops such as spinach [40] or on non-target carbon-related microbial soil processes [41], but further studies are needed to ensure that nitrification inhibitors do not pose any risks to human and environmental health.

It should also be born in mind that this was a pot trial, conducted over only the first five weeks of maize development, so how this result will translate into a field situation over an entire maize growing season can be difficult to extrapolate. However, there is ample evidence for the importance of early maize growing season P supply for yield under field conditions (e.g., [1–5]), so we do consider it likely that our main results will be scalable [42], though probably not one-to-one, to field conditions, and thus could serve as a novel solution for intensive livestock farms.

## 5. Conclusions

Acidification alone increased the water-extractable P fraction of all slurry and digestate organic manures and this promoted an increase in the plant P uptake after the placement application of acidified solid fractions. Acidification alone did not increase P uptake from raw slurry, however when combining acidification with the addition of a nitrification inhibitor, the P uptake significantly increased as well. The acidification treatment combined with a nitrification inhibitor therefore increased the P uptake from all manure products, and this P uptake increase was in line with an increase in plant biomass yield obtained in the treatments receiving the solid fractions.

We conclude that even though the water-soluble P in the organic manures can be increased by acidification alone, an additional beneficial impact on the P uptake is obtained by affecting the N form taken up by the plant, and therefore the combination of acidification and a nitrification inhibitor will result in the greatest likeliness of starter P fertilizing effect of these organic manures. The increase in P fertilizer value of slurry and digestate products through this treatment combination could be sufficient to make them suitable as substitutes of mineral fertilizers applied as starter fertilizers for maize.

**Author Contributions:** Conceptualization and methodology, P.S., J.L., D.M.-S., and L.S.J.; investigation, P.S. and J.L.; data analysis, P.S. and I.R.; writing—original draft preparation, I.R.; writing—review and editing, I.R., P.S., J.L., D.M.-S., and L.S.J.; supervision, D.M.-S. and L.S.J.; funding acquisition, I.R. and L.S.J. All authors have read and agreed to the published version of the manuscript.

**Funding:** This project has received funding from the European Union's Horizon 2020 research and innovation programme under the Marie Sklodowska-Curie grant agreement No 795974, Treat2ReUse.

**Acknowledgments:** Thanks to S. Hvelplund and Måbjerg biogas plant for supplying slurry and digestate materials for the study. Thanks to BASF Demark for providing the Vizura (DMPP) nitrification inhibitor. The lab technical support and help of J.M. Jessen is also greatly acknowledged.

**Conflicts of Interest:** The authors declare no conflict of interest. The funders or providers of research materials had no role in the design of the study; in the collection, analyses, or interpretation of data; in the writing of the manuscript, or in the decision to publish the results.

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
