# Peer review of "Acidified Animal Manure Products Combined with a Nitrification Inhibitor Can Serve as a Starter Fertilizer for Maize"

_agronomy, doi:10.3390/agronomy10121941_

Round 1

Reviewer 1 Report

The study touches upon the topic of enhancing P availability for maize seedlings by acidifying organic fertilizers. The topic is interesting, and the experiment is complete although it is a pot experiment. The manuscript is well written but there are some minor criticisms to address. Please, find these points in the attached file.

Reviewer 2 Report

The manuscript entitled ‚acidified animal manure products combined with a nitrification inhibitor can serve as a starter fertilizer for maize’ fits well to the scope of ‘agronomy’’ and covers highly relevant aspects of agronomic research. Increasing nitrogen and phosphorus efficiency in fertilization is one of the major issues to solved for a climate smart and resource saving management of agronomic systems. The combination of manure acidification and nitrification inhibitor can be considered a novel management option in this domain. The presented pot study can be considered a first step in developing fertilizing strategies based on this innovation. The paper is well written and organized and requires only a minor revision. There are few general points, which should be improved:

  • The novelty of the approach of combining both measures is not clearly enough presented in the introduction as there already exists abundance of knowledge on both acidification and nitrification inhibitors in combination with manure fertilization.
  • The statistical testing ins motivated by the unbalanced experimental design of the treatments. However part of the study, e.g. the manure solid fractions could be analysed more comprehensively by e.g. 3 way Anova (Manure type, acidification, inhibitor). The authors should consider a more comprehensive analysis of at least part of the data set.
  • Why were mainly analysed manure solid fractions? These fractions are rather unlikely to be used as manure injected into the solid close to maize roots of juvenile crop.
  • Potential disadvantages of use of acidification and synthetic nitrification inhibitors should at least be addressed: sulphur oversupply, development of insensibility of soil microbes to inhibitors, (eco)toxicological aspects of (synthetic) inhibitors
  • Calculation of nitrogen and phosphorous balances would largely facilitate the interpretation of the data
  • the paper has still some flaws in use of English language which should be checked and revised

L 19: why was maize used as model crop?

L 97 this hypothesis is formulated rather weakly, as it depends also on the dosage whether one product can replace the other. I assume, the hypothesis implies that same efficiencies can be reached at similar P application rates.

L 152 Please define acronyms in table headings or subscripts

L 269 ff this section is not consistent and somewhat contradictory. Please check and rewrite.

L 414-415 this statement is not very clear, please rephrase and eventually add further information for better understanding

Reviewer 3 Report

The manuscript is well written and clearly presents the study. The information presented is concise yet sufficient, and the results presented are pertinent. I have made some comments for a few improvements, especially in the introduction.

Although both N and P elements were analysed in the manure types, the overall focus of the study is strongly on phosphorus, however this is not reflected neither in the title nor in the abstract. Thus, I suggest to add a sentence in the abstract to clarify the focus on P processes. Also reverse the order of the keywords, to put P first, before N.
L 39: please specify the range of early growth stage of maize referred to.
L 42: specify the negative effects on root development (i.e. presumable “that limit root development” is meant) and also provide a reference
L55: What exactly is meant by “...there is a need for closing the anthropogenic P cycle”? Elaborate this statement in a sentence or two.
L 57: delete the word “actual”
L58: specify which value is being referred to, i.e. the economic or the nutrient value?
L62: Briefly quantify the “positive effects” indicated
L66: this is the first mention of the term “acidification”. Please describe what is meant by this term. Many readers may not be familiar with the treatment or process of manure acidification, i.e. how is manure acidified and why?
L69: change besides to “in addition to”
L79: retaining NH4+ may be important for plant uptake, but it is also important to retain NO3- in the upper soil layers, to avoid or reduce leaching. Can this point be addressed here as well?
L 85: delete the word “productivity”
L101: insert a sentence or two at the beginning of this section that states the different types of manures used upfront and explain them briefly, i.e. In this study, RS, RSSF and DSF were tested....
L139: the pots were 10 cm in diameter with 3 corn plants growing in them. Was this not too crowded? This does not represent field conditions, so from a practical perspective it would be useful and interesting to state the N applied to each pot in kg/ha and how this compares to field conditions? This information can also go in the discussion if it is better suited.
L 177-179: The pot water holding capacity is not so relevant. More important is the water filled pore space and if the irrigation was at or above field capacity. It is stated that 26% of the dry soil weight was water, which means 0.65 L was water. From the information on L 139, the dry bulk density appears to be 1.14 g/cm3. Thus, is it possible to make inferences about the WFPS? With the particle density this could be calculated.
Fig 2 is missing!
L319: the information is not easy to follow. I recommend to use consistent terminology throughout the paper. Please specify what treatments are meant by “solid fractions from raw slurry”, is this the RSSF, and which is the “digestate”, is this the DSF?
L322: here also please specify the acronyms in brackets for example, of the acidified manure treatments that are referred to.
L327: also not clear or easy to follow and cannot relate the information to Fig 8. Specify which treatments are meant by “the same non-treatment manures”.
L326-327: will the potential for nitrate leaching be decreased in the aniRSSF or aniDSF treatments? This can also go into the discussion section.
L389: correct “occurred” to “occur”
L424: I believe the word “treatments” is mean to be used instead of “samples”.
L425: replace the word “born” with “kept”
In the discussion I would like to have seen some information pertaining to L121-122, namely how practical is the acidification process is for farmers to undertake on their farms? Or would they purchase the manure in an already acidified state?

Author Response

This manuscript is a resubmission of an earlier submission. The following is a list of the peer review reports and author responses from that submission.

Round 1

Reviewer 1 Report

The manuscript ‚Acidified animal manure products combined with a nitrification inhibitor can serve as a starter fertilizer for maize’ fits well into the scope of agronomy. To ensure and increase the nutrient use efficiency of organic manure and the connected possible reduction of total nutrient supply to crops and losses to the environment is of pivotal for agro environmental policies in many countries.
In the presented pot trial, a juvenile maize crop was analysed for the effects of slurry acidification and/or the addition of nitrification inhibitor DMPP on nutrient uptake and crop growth until 35 DAS. Technically the trial is of high standard but the trial design is unbalanced:
- Only cattle raw slurry and no raw digestates
- Different amounts of sulfuric acid and thereby variable sulphur supply for the crops
- Unbalanced N supply
- Unbalanced DMPP supply as adjusted to the N supply
This unbalance should be considered in an appropriate statistical analysis. The use of a one factorial ANOVA is too crude and can result in misinterpretation of the results. The data should be analysed by multifactorial models, e.g. two way ANOVA for the cattle slurry types. The whole data set could also be investigated by an ANCOVA approach involving actual data for N, S and DMPP supply. The whole statistical anlysis should be revised and the used statistical models be presented in more detail.
In addition, the soil amount used in this study was rather small, providing only very limited resource for the crops to explore for nutrients. This may have resulted in stronger effects as compared to field conditions. This should be considered in the discussion and in the derivation of conclusions.
In the introduction the authors should describe in more detail the novelty of the research question.
The overall writing of the manuscript is of high quality although there are some grammatical flaws. So, the authors should check the language again.

Points in detail:
L 23 give name of inhibitor used
L 25-30 please give some figures on the observed effects
L 50-57 this paragraph can be shortened. The last two sentences can be combined in one sentence.
L 94 give full description of WEP as the acronym has not been presented before.
L 107 why was only the solid fraction of the digestate used in this study? Please explain.
L 160 Was the soil recompacted? Please consider soil conditions and amount and also temperature regime applied in the growth chamber in the interpretation of the rusults.
L 175 us ‘analyses’ in place of ‘analysis’
Fig 1: symbols are hard to distinguish, please chose bigger or other symbols.
Fig 2. Why are no data presented for NI treatment?
L 310 discussion: much more attention should be given to unbalanced trial design and limitations of pot trials with respect to the extrapolation of results to field conditions.

Reviewer 2 Report

Dear Authors, 

I liked your paper. It is well written and easy to follow. Although the experiment was a lab one, it represents a preliminary step to gain insights about the interaction of NI (i.e., Vizura) and treated slurry (i.e., acidification).

Please, solve the following minor flaws:

Lines 80-82: Please, add some reference. E.g., recently, Chiodini et al. have published on Agronomy a study on the benefits of the addition of NI (Vizura).
Chiodini, M.E., Perego, A., Carozzi, M., Acutis, M., 2019. The nitrification inhibitor Vizura® Reduces N2O emissions when added to digestate before injection under irrigated maize in the Po valley (Northern Italy). Agronomy, 9(8), 431. ISSN: 20734395. DOI: 10.3390/agronomy9080431

Line 94: WEP to be replaced with water-extractable P

Line 200: is the ANOVA a 1-way ANOVA? I guess there is one only factor in the model, with 15 levels. Plus, in the Material and Methods you should add the number of replications. 

Figures: in the figures (from 3 on) you should add the explanation of the acronymous in the caption because they are too many to recall. Also, How did you assign letters to levels (considering the 15 levels together or splitting them into four groups)? It seems that the 15 levels are split into four groups. Please, build the graphs according to the statistical analysis which you performed.